# Analysis of Sphingolipids in Pediatric Patients with Cholelithiasis—A Preliminary Study

**DOI:** 10.3390/jcm11195613

**Published:** 2022-09-23

**Authors:** Katarzyna Zdanowicz, Anna Bobrus-Chcociej, Karolina Pogodzinska, Agnieszka Blachnio-Zabielska, Beata Zelazowska-Rutkowska, Dariusz Marek Lebensztejn, Urszula Daniluk

**Affiliations:** 1Department of Pediatrics, Gastroenterology, Hepatology, Nutrition and Allergology, Medical University of Bialystok, 15-274 Bialystok, Poland; 2Department of Hygiene, Epidemiology and Metabolic Disorders, Medical University of Bialystok, 15-22 Bialystok, Poland; 3Department of Pediatric Laboratory Diagnostics, Medical University of Bialystok, 15-274 Bialystok, Poland

**Keywords:** cholelithiasis, gallstones, sphingolipids, children

## Abstract

(1) Background: Disturbances in the sphingolipid profile are observed in many diseases. There are currently no data available on the evaluation of sphingolipids and ceramides in cholelithiasis in children. The aim of this study was to evaluate the concentrations of sphingolipids in the sera of pediatric patients with gallstones. We determined their relationship with anthropometric and biochemical parameters. (2) Methods: The concentrations of sphingolipids in serum samples were evaluated using a quantitative method, ultra-high-performance liquid chromatography–tandem mass spectrometry. (3) Results: The prospective study included 48 children and adolescents diagnosed with gallstones and 38 controls. Serum concentrations of total cholesterol (TC); sphinganine (SPA); ceramides—C14:0-Cer, C16:0-Cer, C18:1-Cer, C18:0-Cer, C20:0-Cer and C24:1-Cer; and lactosylceramides—C16:0-LacCer, C18:0-LacCer, C18:1-LacCer, C24:0-LacCer and C24:1-LacCer differed significantly between patients with cholelithiasis and without cholelithiasis. After adjusting for age, gender, obesity and TC and TG levels, we found the best differentiating sphingolipids for cholelithiasis in the form of decreased SPA, C14:0-Cer, C16:0-Cer, C24:1-LacCer and C24:0-LacCer concentration and increased C20:0-Cer, C24:1-Cer, C16:0-LacCer and C18:1-LacCer. The highest area under the curve (AUC), specificity and sensitivity were determined for C16:0-Cer with cholelithiasis diagnosis. (4) Conclusions: Our results suggest that serum sphingolipids may be potential biomarkers in pediatric patients with cholelithiasis.

## 1. Introduction

The incidence of cholelithiasis is constantly increasing in the pediatric population, ranging from 1.9% to 4% [1,2]. Factors that predispose to the development of cholelithiasis include hemolytic disorders, ceftriaxone therapy, total parenteral nutrition (TPN), cystic fibrosis, obesity, genetic predisposition and hipercholesterolemia [3]. However, obesity is not only a growing problem in the pediatric population worldwide, but is also the most common cause of cholelithiasis in children today [1,4]. Additionally, similarly to adults, a significantly greater incidence of gallstone disease is observed in girls than in boys [5]. The formation of cholesterol gallstones is connected with hypersecretion of cholesterol and supersaturated bile in the gallbladder. Despite the known predisposing factors, such as genes and lifestyle, the mechanism of this disease has not been clearly defined [6,7].

Novel omics techniques such as metabolomics and lipidomics are promising tools that allow the measurement of carbohydrates, lipids, amino acids, amines or steroids. Sphingolipids are a class of bioactive lipids that include, i.a., ceramides (CER), lactosylceramides (LacCer), sphingosine (Sph) and sphinganine (SPA). The roles of these lipids are complex and involve the inflammatory response and cellular metabolism, migration and signaling. CER may also be involved in growth and apoptosis [8]. Several studies have found that sphingolipids may be associated with obesity, insulin resistance, inflammation-related illnesses and cancer [9,10]. To the best of our knowledge, there are currently no data available on the evaluation of sphingolipids and ceramides in cholelithiasis in both children and adults. Changes in the concentrations of adipokines may be indirect indicators of lipid disturbances. In the studies published so far in children, the effect of chemerin on the presence of cholelithiasis has been observed [11]. Chemerin, as an adipokine involved in adipogenesis, glucose homeostasis and energy metabolism [12], may be associated with lipid disturbances.

The aim of our study was to analyze the concentrations of sphingolipids, including C16:0-LacCer, C18:0-LacCer, C18:1-LacCer, C24:0-LacCer, C24:1-LacCer, C14:0-Cer, C14:0-Cer, C16:0-Cer, C18:0-Cer, C18:1-Cer, C20:0-Cer, C22:0-Cer, C24:0-Cer, C24:1-Cer, SPA and Sph, in the sera of pediatric patients with gallstones to broaden the knowledge of the pathogenesis of this disease. Moreover, we analyzed the relationships between the concentrations of the sphingolipids and anthropometric measurements and biochemical lipid profile parameters.

## 2. Materials and Methods

### 2.1. The Study Groups

This prospective study involved children with an initial diagnosis of cholelithiasis who were admitted to our Department from January 2017 to December 2018. They were diagnosed by abdominal ultrasound. None of the patients were treated with ursodeoxycholic acid at study entry. Patients included in the study group had no stone formation in other organs confirmed by abdominal ultrasound or magnetic resonance cholangiopancreatography (MRCP). In their family histories there were no diseases predisposing to cholelithiasis (e.g., cystic fibrosis). The family history was irrelevant, since only a few patients reported a positive family history of cholelithiasis. All patients had body mass index (BMI) calculated based on the World Health Organization; children were overweight or obese if their BMIs were ≥85th percentile. The control group included 38 children without any somatic organ pathology. Patients were excluded from the analysis if they were diagnosed with complications of cholelithiasis (for example, gallstone pancreatitis), bile duct defects, hemolytic or infectious diseases. None of the patients were taking medications that affect lipid or carbohydrate metabolism.

Written informed consent was obtained from the parents of all the study participants. The protocol was approved by the local Bioethics Committee prior to patient recruitment, and the study was in accordance with the Helsinki Accords (approval number: R-I-002/393/2016, APK.002.464.2020).

### 2.2. Collection of Samples for Analysis

All participants underwent a physical examination. Blood samples were taken from all participants after a 10 h overnight fast and were immediately centrifuged and frozen at −80 °C for further analysis. Total cholesterol (TC) and triglycerides (TG) levels were measured by a homogenous enzymatic colorimetric method, and values < 5.17 mmol/L for TC and 0.4–1.69 mmol/L for TG were considered normal. 

### 2.3. Measurement of Sphingolipids 

The content of sphingolipids was measured using ultra-high performance liquid chromatography–tandem mass spectrometry (UHPLC/MS/MS) according to Bielawski et al. [13]. An internal standard mixture (Sph-d7, SPA-d7, C15:0-d7-Cer, C16:0-d7-Cer, C18:1-d7-Cer, C18:0-d7-Cer, 17C20:0-Cer, C24:1-d7-Cer, C24-d7-Cer, C16-LacCer, C17-LacCe, C18-LacCer) (Avanti Polar Lipids, Alabaster, Al, USA) and an extraction mixture (isopropanol:ethyl acetate, 15:85; *v*:*v*) were added to each sample (100 uL of serum). The mixture was vortexed, sonicated and then centrifuged for 5 min at 3000× *g*, 4 °C. The supernatant was transferred to a new tube, and the pellet was re-extracted. After centrifugation, the supernatants were combined and evaporated under nitrogen. The dried sample was reconstituted in 100 μL of LC Solvent B B (2 mM ammonium formate, 0.15% formic acid in methanol) or LC/MS/MS analysis. Sphingolipids were analyzed using a Sciex Qtrap 6500 + triple quadrupole mass spectrometer (SCIEX, Framingham, MA, USA) using a positive ion electrospray ionization (ESI) source (except S1P, which was analyzed in negative mode) with multiple reaction monitoring (MRM), against standard curves constructed for each analyzed compound. The chromatographic separation was performed on a reverse-phase Zorbax SB-C8 column, 2.1 × 150 mm, 1.8 um (Agilent Technologies, Santa Clara, CA, USA), in a binary gradient using 1 mM ammonium formate and 0.1% formic acid in water as solvent A, and 2 mM ammonium formate and 0.1% formic acid in methanol as solvent B at the flow rate of 0.4 mL/min.

### 2.4. Statistical Analysis

Comprehensive data were processed using IBM SPSS Statistics 25.0 (Chicago, IL, USA) and shown as the median, minimum and maximum values. Statistical analysis was performed using the Mann–Whitney test for quantities and the chi-square test for categorical variables. Spearman’s correlation test was used to analyze the correlations between variables. A generalized multivariable linear model was created to determine the association between ceramides levels and the presence of cholelithiasis after adjusting for presence of age, obesity, sex and levels of TG and TC. Receiver operating characteristic (ROC) curves were generated by using the presence of cholelithiasis as a classification variable and concentrations of sphingolipids as prognostic variables (data were analyzed using Statistica 13.3 package, TIBCO Software Inc., Cracow, Poland). Statistical significance was considered to be met when the *p* value was less than 0.05.

## 3. Results

The prospective study included 48 children and adolescents diagnosed with gallstones and 38 controls. Among patients with cholelithiasis, 21 (43.75%) were overweight/obese, and 27 (56.25%) had normal BMIs, whereas 15 (39.47%) were overweight/obese and 23 (60.53%) had normal BMIs in the control group. The demographic data and laboratory results of each group are presented in Table 1. All subjects included in the study were Caucasian. 

### 3.1. Sphingolipid Concentration in Study Group and Control Group

Serum concentrations of TC, SPA, C14:0-Cer, C16:0-Cer, C18:1-Cer, C18:0-Cer, C20:0-Cer, C24:1-Cer, C16:0-LacCer, C18:0-LacCer, C18:1-LacCer, C24:0-LacCer and C24:1-LacCer differed significantly between patients with cholelithiasis and without cholelithiasis (Table 1).

### 3.2. Correlation between Sphingolipids, Lipid Profile and BMI

The correlations between selected sphingolipids (with the greatest prognostic significance for cholelithiasis) and BMI, TC and TG are summarized in Table 2. The following significant correlations of ceramides with BMI were noted: C16:0-Cer, C12:0-Cer and C24:1-Cer; significant correlations with TG were found for C14:0-Cer, C24:1-Cer and C24:0-LacCer; and significant correlations for TG were determined for C14:0-Cer, C16:0-Cer, C18:1-LacCer, C24:1-LacCer and C24:0-LacCer.

The ROC analysis presented in Table 3 and Appendix A was performed to determine which sphingolipids had the best predictive value for distinguishing children with cholelithiasis from those without gallstones. The best result was determined for C16:0-Cer. A cut-off value of 59.692 ng/mL discriminated children with cholelithiasis with 97.9% sensitivity and 100% specificity (AUC = 1.0). Additionally, a high AUC value (0.99) was obtained for C14:0-Cer at the cut-off value 1.363 ng/mL with 95.8% sensitivity and 97.4% specificity.

### 3.3. Generalized Multivariable Linear Model

After adjusting for age, gender, obesity and TC and TG levels, we found the best differentiating sphingolipids for cholelithiasis in the form of decreased SPA, C14:0-Cer, C16:0-Cer, C24:1-LacCer and C24:0-LacCer concentrations; and increased C20:0-Cer, C24:1-Cer, C16:0-LacCer and C18:1-LacCer (Table 4).

## 4. Discussion

To our knowledge, this is the first report on sphingolipid analysis in pediatric patients with cholelithiasis. The available data on the role of ceramides in gallstone disease were obtained in animal models. In two animal studies, inhibition of ceramide biosynthesis by myriocin suppressed gallstone formation [14,15]. Myriocin is a natural inhibitor of serine palmitoyltransferase, an enzyme involved in the initial synthesis of sphingolipids [16]. This observation may suggest that sphingolipids are involved in gallstone pathogenesis. Unfortunately, the ceramide profile was not analyzed in this study. In another study in genetically engineered mice on a lithogenic diet, cholesterol gallstone formation was observed in 70% of cases, as opposed to 40% of mice on the same diet with supplementation of myrocin. At the same time, significant increases in the serum and bile ceramide concentrations were observed in mice on a lithogenic diet. In the group additionally treated with myrocin, the levels of ceramides were lower [15]. According to the authors, it is not known whether increases in serum and bile ceramides were directly related to the formation of gallstones; unfortunately, the ceramide profile was not analyzed in this study. In addition, in the group without myrocin and with increased serum ceramides also showed increased liver and ileum expression of ABCG5 and ABCG8 mRNA—i.e., the ATP binding cassette (ABC) transporter was noted. The ABC polymorphism has been reported to increase the risk of gallstone formation through the efflux of cholesterol in the liver [17]. Another factor that may link sphingolipids with gallstone formation is the presence of alkaline sphingomyelinase in human bile and liver, an enzyme that hydrolyzes sphingomyelin to ceramide in a bile-salts-dependent manner [18].

In our study, in patients with cholelithiasis, regardless of age, body weight, gender and levels of TC and TG, significant differences were observed in concentrations of SPA, C14:0-Cer, C16:0-Cer, C20:0-Cer, C24:1-Cer, C16:0-LacCer, C18:1-LacCer, C24:1-LacCer and C24:0-LacCer compared with children without gallstones. Our analysis showed that the most sensitive and specific markers of cholelithiasis among ceramides were decreased C14:0-Cer and C16:0-Cer. Regarding results for other diseases in children, elevated levels of ceramides C14:0-Cer and C16:0-Cer have been observed in non-alcoholic fatty liver disease (NAFLD) [19]. Moreover, Chang et al. described a bidirectional association between occurrence of NAFLD and cholelithiasis [20]. However, the mechanism of the coexistence of these two diseases has not been clearly identified. It is likely that insulin resistance, which increases de novo lipogenesis and activation of bile acid signaling pathways, may be involved in this process [21]. Due to the possible influence of circulating sphingolipids, patients with NAFLD were not included in our study. Maldonado-Hernández et al. also found higher levels of C14:0-Cer in adolescents with hepatic steatosis (HS). C14:0-Cer significantly correlated with total cholesterol in obese patients with HS [22]. Impaired cholesterol metabolism plays an important role in gallstone formation. In children with cholesterol gallstones, low intestinal cholesterol absorption was measured by increased serum plant sterols [23]. The accumulation of cholesterol and plant sterols in gallstones mirrors increased liver secretion of cholesterol [24]. We observed a significantly lower concentration of TC in children with cholelithiasis, regardless of body weight. However, we did not determine cholesterol metabolites and phytosterols. In our study, the composition of gallstones was not specified.

We have also demonstrated that a decrease in C24:0-LacCer or C24:1-LacCer, or an increase in C18:1-LacCer, may predispose one to cholelithiasis. In pediatric patients with type 1 diabetes, lower levels of C18-LacCer and C24-LacCer were associated with an increased risk of progression in chronic kidney disease [25]. In other study conducted in adults, lactosylceramides were not associated with diabetes [26]. On the other hand, a higher amount of liver C24:0-LacCer was observed in patients with non-alcoholic steatohepatitis (NASH) than in the controls [27]. The above data may suggest that, depending on the disease, the concentrations of individual lactosylceramides may have protective or aggravating effects on the course of the disease. 

Metabolic syndrome is a potential risk factor for the increased occurrence of gallstones [28]. However, it is not known whether gallstones are a determinant of the development of metabolic syndrome and insulin resistance. Type 2 diabetes is one of the important risk factors for the development of gallstone disease in adult patients [29]. In our study, we found elevated C20:0-Cer levels in patients with cholelithiasis, but none of our patients had been diagnosed with metabolic syndrome at the time of evaluation. In adult studies, C20:0-Cer levels were higher in obese patients with type 2 diabetes mellitus; unfortunately, in this study there were no data on the coexistence of gallstone disease [30]. The plasma level of C20:0-Cer was higher not only in patients with type 2 diabetes mellitus, but also 3 years before diagnosis [31]. It would be interesting to see if patients already have abnormal level of C20:0-Cer or other ceramides prior to development of cholelithiasis. In an adult study it was reported that the risk of type 2 diabetes could be predicted using concentrations of C16:0-Cer, C18:0-Cer and C18:1-Cer. In addition, the loss of body weight is accompanied by significant reductions in the ceramide index and the risk of diabetes [32]. However, more research is needed to assess the effects of selected sphingolipid levels on the incidence of diabetes and gallstone development. Dietary fat consumption is also a risk factor for the development of cholesterol gallstones. In animal models, a high-fat diet was associated with increases in body weight, fasting glucose, and insulin levels. Moreover, the imbalance in glucose metabolism resulted in increased levels of C18:1-Cer, C18:0-Cer, C24:1-Cer and C24:0-Cer in submandibular gland cells [33]. In other studies, a relationship has also been observed between the increased dietary intake of saturated fats and the serum level of ceramides [34,35]. Ceramides are cholesterol-independent biomarkers of an overlapping but distinct spectrum of diseases [35]. Zabielski et al. also found higher concentrations of C20: 0-Cer in serum and liver in rats on a high-fat diet [36]. 

Other studies also attempted to assess the usefulness of ceramides in the courses of various diseases. Interestingly, in our study we found increased serum C16:0-LacCer in children with cholelithiasis independently of the other risk factors, such as overweight/obesity and gender. The same lactosylceramide was found to be very specific to children with Crohn’s disease, a type of inflammatory bowel disease [9]. It would be interesting to investigate whether the ileal expression of ABCG5 and ABCG8 is associated with this inflammatory bowel disease, which has as-yet-unknown pathogenesis, and to evaluate the effect of myrocin treatment on the course of intestinal inflammation in an animal model of inflammatory bowel disease.

This is the first study in pediatric patients with cholelithiasis to investigate serum sphingolipids. The novelty of these findings is the main strength of our study and may be an introduction to further analyses in this age group. However, our work has several potential limitations. First, the small number of participants did not allow us to generalize our results. Our study was designed as an explanatory study to generate only pathophysiological theories of disease. The number of patients enrolled in the study was low due to the time span and the monocentric nature of the study. We understand that our results may be subject to errors of omission (type II error), and we did not interpret non-significant statistical results as underlying a true lack of differences. We also did not analyze the eating habits, physical activity and insulin resistance, which may also impact the results.

## 5. Conclusions

Our results suggest that serum sphingolipids are potential biomarkers for patients with cholelithiasis. However, it is important to conduct further research and answer the question of whether sphingolipids are factors regulating the formation of gallstones, or changes in their concentration are only secondary effects of cholelithiasis. It would be interesting to see if patients already have abnormal level of ceramides prior to the development of cholelithiasis.

## Figures and Tables

**Table 1 jcm-11-05613-t001:** Comparative characteristics of patients with and without cholelithiasis.

Marker	Patients with Cholelithiasis	Patients without Cholelithiasis	*p*
Number of patients	48	38	NA
Age (median, range)	12 (1–17)	12 (4–17)	NS
Sex (male)	21	22	NS
BMI (kg/m^2^)	20.45 (13.3–29)	21.82 (13.3–39.54)	NS
BMI z-score (median, range)	0.98 (−1.73–2.59)	1.17 (−2.09–2.9)	NS
TC (mmol/L)	3.8 (2.41–6.54)	4.16 (3.21–7.71)	0.04
TG (mmol/L)	1.73 (0.37–1.96)	0.94 (0.43–2.07)	NS
Sph (ng/mL)	2.35 (1.44–3.21)	2.0 (1.37–3.23)	NS
SPA (ng/mL)	1.40 (1.11–2.23)	1.69 (1.03–2.66)	0.006
C14:0-Cer (ng/mL)	0.94 (0.44–1.51)	1.947 (1.01–2.52)	<0.001
C16:0-Cer (ng/mL)	42.48 (27.42–96.93)	85.24 (61.20–125.66)	<0.001
C18:1-Cer (ng/mL)	1.45 (1.04–2.18)	1.90 (0.92–4.44)	0.01
C18:0-Cer (ng/mL)	220.25 (125.50–318.55)	273.91 (162.28–438.99)	0.006
C20:0-Cer (ng/mL)	38.67 (23.47–58.97)	15.92 (10.73–39.07)	<0.001
C22:0-Cer (ng/mL)	221.48 (158.69–286.96)	235.12 (153.22–498.21)	NS
C24:1-Cer (ng/mL)	471.50 (335.72–651.07)	395.00 (245.40–641.9)	0.01
C24:0-Cer (ng/mL)	725.99 (556.03–941.12)	759.68 (469.39–1164.63)	NS
C16:0-LacCer (ng/mL)	1524.09 (901.77–2067.22)	1356.47 (876.70–1725.39)	<0.001
C18:0-LacCer (ng/mL)	77.16 (60.53–116.27)	85.04 (62.72–116.88)	0.01
C18:1-LacCer (ng/mL)	26.44 (19.00–39.96)	19.45 (12.88–28.10)	<0.001
C24:1-LacCer (ng/mL)	610.90 (423.31–827.57)	857.66 (552.53–1134.32)	<0.001
C24:0-LacCer (ng/mL)	188.33 (134.33–284.36)	277.38 (198.05–428.39)	<0.001

ALT—alanine transaminase; BMI- body mass index, GGT—gamma-glutamyltransferase; TC—total cholesterol; TG—triglycerides; Sph—sphingosine; SPA—sfinganine; Cer—ceramide; LacCer—lactosylceramide; NS—not significant.

**Table 2 jcm-11-05613-t002:** The correlations of sphingolipids with BMI, TC and TG in children with cholelithiasis.

	BMI	TG	TC
SPA	NS	NS	NS
C14:0-Cer	NS	R = 0.29; *p* = 0.01	R = 0.55; *p* < 0.001
C16:0-Cer	R = 0.28; *p* = 0.01	NS	R = 0.32; *p* = 0.003
C18:1-Cer	NS	NS	NS
C18:0-Cer	NS	NS	NS
C20:0-Cer	R = −0.35; *p* = 0.002	NS	NS
C24:1-Cer	R = −0.3; *p* = 0.007	R = 0.37; *p* < 0.001	NS
C16:0-LacCer	NS	NS	NS
C18:1-LacCer	NS	NS	R = −0.28; *p* = 0.01
C18:0-LacCer	NS	NS	NS
C24:1-LacCer	NS	NS	R = 0.24; *p* = 0.03
C24:0-LacCer	NS	R = 0.24; *p* = 0.03	R = 0.29; *p* = 0.009

SPA—sfinganine; BMI—body mass index, Cer—ceramide, LacCer—lactosylceramide, TG—triglicerides, TC—total cholesterol, NS—not significant.

**Table 3 jcm-11-05613-t003:** Analysis of the diagnostic efficiency of selected sphingolipids that significantly differentiated patients with and without cholelithiasis.

Marker	AUC	95% C.I. AUC	*p*	Cut-Off	Sensit.	Spec.	ACC
SPA	0.326	(0.211–0.442)	0.003	1.107	100%	26%	57.0%
C14:0-Cer	0.99	(0.971–1.0)	<0.001	1.363	95.8%	97.4%	96.5%
C16:0-Cer	1	(1.0–1.0)	<0.001	59.692	97.9%	100%	98.8%
C20:0-Cer	0.914	(0.857–0.972)	<0.001	26.83	93.8%	73.7%	84.9%
C24:1-Cer	0.656	(0.533–0.778)	0.01	406.594	77.1%	60.5%	69.8%
C16:0-LacCer	0.721	(0.615–0.827)	<0.001	1536.383	50%	92.1%	68.6%
C18:1-LacCer	0.888	(0.819–0.957)	<0.001	21.977	89.6%	78.9%	84.9%
C24:1-LacCer	0.876	(0.804–0.948)	<0.001	749.27	65.8%	93.8%	81.4%
C24:0-LacCer	0.93	(0.88–0.979)	<0.001	250.785	73.7%	95.8%	86.0%

SPA—sfinganine; Cer—ceramide, LacCer—lactosylceramide AUC—area under the curve; C.I-confidence interval; Sensit.—sensitivity; Specific.—specificity; ACC—accuracy.

**Table 4 jcm-11-05613-t004:** Effects of cholelithiasis diagnosis on ceramide concentrations after adjusting for age, obesity, sex and levels of TG and TC (generalized multivariable linear model).

Dependent Variables	Model Coefficient (*β*)	*p*
Sph	0.001	NS
SPA	−0.373	<0.0001
C14:0-Cer	−1.014	<0.0001
C16:0-Cer	−52.120	<0.0001
C18:1-Cer	0.045	NS
C18:0-Cer	−2.545	NS
C20:0-Cer	19.839	<0.0001
C22:0-Cer	17.868	NS
C24:1-Cer	83.609	<0.0001
C24:0-Cer	55.559	NS
C16:0-LacCer	177.932	0.01
C18:0-LacCer	−7.504	NS
C18:1-LacCer	8.419	<0.0001
C24:1-LacCer	−252.245	<0.0001
C24:0-LacCer	−84.788	<0.0001

Sph—sphingosine, SPA—sfinganine; Cer—ceramide, LacCer—lactosylceramide, TG—triglicerides, TC—total cholesterol, NS—not significant.

## Data Availability

The datasets generated and analyzed during the current study are available from the corresponding author on reasonable request.

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
