# Peer review of "Analysis of Sphingolipids in Pediatric Patients with Cholelithiasis—A Preliminary Study"

_jcm, 2022, doi:10.3390/jcm11195613_

Round 1

Reviewer 1 Report

The study is simple but clearly presented, that the authors evaluated the concentrations of sphingolipids in the serum of pediatric patients with gallstones via HPLC-MS-MS and found certain serum sphingolipids, e.g., total cholesterol, sphinganine, ceramides tosyceramides were differed significantly between patients with and without cholelithiasis. So the authors suggested serum sphingolipids could be applied as biomarkes in pediatric patients with cholelithiasis.

There are still some suggestions for the authors:

1. the sample size is relative small, could you get more patients to validate your premiary finding, even patients without cholelithiasis to exclude those irrelevant serum lipids?

2. Which serum lipid or what kind of combination of lipid is most powerful for prediction of cholelithiasis?

3. Do these patients with cholelithiasis have stones in other organs, kidney or intrahepatic biliary?  the serum sphingolipids are also related to stones formatoin in other parts?

Author Response

Thank You for the evaluation of our manuscript. We are very grateful for Your valuable comments.

Please find our answers below.

There are still some suggestions for the authors:

  1. the sample size is relative small, could you get more patients to validate your premiary finding, even patients without cholelithiasis to exclude those irrelevant serum lipids?

A: Patients included in the control group were appropriately selected in terms of demographic data. Additionally, the determination of the lipid level was performed in all patients at the same time to exclude a possible laboratory error. Expanding the research and control groups at this point in time may lead to false results and conclusions. The small number of patients included in the study is a limitation of our study, which was emphasized in the article.

  1. Which serum lipid or what kind of combination of lipid is most powerful for prediction of cholelithiasis?

Answer: We have completed the text as given below:

Line 146-150: The ROC analysis presented in Table 3 was performed to determine which sphingolipids had the best predictive value for distinguishing children with cholelithiasis from those without gallstones. The best result was determined for C16:0-Cer. A cut-off value of 59.697 ng/ml discriminated children with cholelithiasis with 97.9% sensitivity and 100% specificity (AUC = 1.0).

  1. Do these patients with cholelithiasis have stones in other organs, kidney or intrahepatic biliary? the serum sphingolipids are also related to stones formation in other parts?

Answer: We have completed the text as given below:

Line 70-72: Patients included in the study group had no stone formation in other organs confirmed by abdominal ultrasound or magnetic resonance cholangiopancreatography (MRCP) .

Reviewer 2 Report

Zdanowicz et al. investigated the serum sphingolipid concentrations in pediatric patients with cholelithiasis. Despite some issues, this study is interesting and offers new information on the relationship between serum sphingolipid concentrations and pediatric cholelithiasis.

  1. The number of cases and controls was limited, and the sample size was the main cause for concern. Please describe the procedures for calculating statistical power and sample size.
  2. Population: It would be helpful if a few details were clarified. Please provide information on the study population's race and ethnicity and adjust race/ethnicity in the multivariable analysis. Did the study participants have a family history of cholelithiasis and cystic fibrosis? Was this the study participant's initial cholelithiasis diagnosis? 
  3. Methods: It was unclear which factors were adjusted in the multivariable models. It was mentioned in the text on line 50 that the authors adjusted for age, however, other places (Table 4 and Discussion line 8) did not indicate this. Was the age of the study participants adjusted? 
  4. Methods: What definition of obesity did the authors employ for their analyses? Age and sex-specific BMI categories differ from adult BMI categories for children and adolescents. The definition of obesity used for the study might be helpful to readers.
  5. Results: Please describe how the optimal cut-off value was determined in the ROC analysis. Specifically, the upper range of C16:0-Cer in study subjects is 59.69 ng/ml (Table 1), but the AUC of C16:0-Cer is 1 with a cut-off value of 59.697 ng/ml. I suggest providing ROC curves of statistically significant plasma sphingolipids to improve comprehension. 
  6. Does the sphingolipid profile differ by number, type of gall stones, or disease severity?  
  7. Were there any gaps in time between the diagnosis of cholelithiasis and the evaluation of physical examinations and lab results? Can a difference in sphingolipid value over time influence the relationship between sphingolipid and cholelithiasis?

Minor 

  • The decimal separators in Table 1 are not consistent. 
  • Please mention whether all the variables, including each sphingolipid, were available for the study participants or whether there were any missing data. 

Author Response

Thank You for the evaluation of our manuscript. We are very grateful for Your valuable comments.

Please find our answers below.

The number of cases and controls was limited, and the sample size was the main cause for concern. Please describe the procedures for calculating statistical power and sample size.

Answer: Our study included all children diagnosed with cholelithiasis and hospitalized in our department in years 2017 – 2018. The incidence of gallstone disease in children is much lower than in adults, and because it is a factor beyond our control, the number of patients included in the study was the maximum for our center. As we explained in the discussion part (line: 259-264), the number of our study group is limitation of the study. However, this is the only study already performed worldwide based on our knowledge.

Population: It would be helpful if a few details were clarified. Please provide information on the study population's race and ethnicity and adjust race/ethnicity in the multivariable analysis. Did the study participants have a family history of cholelithiasis and cystic fibrosis? Was this the study participant's initial cholelithiasis diagnosis?

Answer: We have completed the text as given below:

Line 124: All subjects included to the study were Caucasian.

Line 70-74: In patients included in the study group, no stone formation was found in other organs, which was confirmed by abdominal ultrasound or magnetic resonance cholangiopancreatography (MRCP). In family history there were no diseases predisposing to cholelithiasis (e.g. cystic fibrosis). The family history was irrelevant, since only few patients reported positive family history of cholelithiasis.

Line 67-68: This prospective study involved a group of 48 pediatric patients with initial diagnosis of cholelithiasis admitted to our Department from January 2017 to December 2018

Methods: It was unclear which factors were adjusted in the multivariable models. It was mentioned in the text on line 50 that the authors adjusted for age, however, other places (Table 4 and Discussion line 8) did not indicate this. Was the age of the study participants adjusted?

Answer: Age was adjusted. Missing data were completed in Table 4 and Discussion.

Methods: What definition of obesity did the authors employ for their analyses? Age and sex-specific BMI categories differ from adult BMI categories for children and adolescents. The definition of obesity used for the study might be helpful to readers.

Answer: We have completed the text as given below:

Line 74-76: All patients had calculated body mass index (BMI) based on the World Health Organi-zation, children were overweight or obese if their BMI were ≥85th percentile

Results: Please describe how the optimal cut-off value was determined in the ROC analysis. Specifically, the upper range of C16:0-Cer in study subjects is 59.69 ng/ml (Table 1), but the AUC of C16:0-Cer is 1 with a cut-off value of 59.697 ng/ml. I suggest providing ROC curves of statistically significant plasma sphingolipids to improve comprehension.

Answer:

The data has been corrected in Table 1

Line 115-118: Receiver operating characteristic (ROC) curves were generated using presence of cholelithiasis as a classification variable and concentrations of sphingolipids as prognostic variables (data were analyzed using Statistica 13.3 package, TIBCO Software Inc., Cracow, Poland).

Answer: We have completed the text as given below:

Line 147-151: The ROC analysis presented in Table 3 and Figure 1 Supplementary Materials was performed to determine which sphingolipids had the best predictive value for distinguishing children with cholelithiasis from those without gallstones.

Does the sphingolipid profile differ by number, type of gall stones, or disease severity? 

Answer: As mentioned in the text (line 77-79), only patients with cholesterol stones and no serious complications of the disease were enrolled in the study. It would be difficult to count the number of stones in the gallbladder because some patients had a conglomerate of stones, so we omitted this assessment.

Line 77-79: Patients were excluded from the analysis if they were diagnosed with complications of cholelithiasis (for example gallstone pancreatitis), bile duct defects, hemolytic and infectious diseases.

Were there any gaps in time between the diagnosis of cholelithiasis and the evaluation of physical examinations and lab results? Can a difference in sphingolipid value over time influence the relationship between sphingolipid and cholelithiasis?

Answer: Diagnosis, demographic data and laboratory results were obtained at the same time.

Minor

The decimal separators in Table 1 are not consistent.

Answer: It has been corrected.

Please mention whether all the variables, including each sphingolipid, were available for the study participants or whether there were any missing data.

All demographic data and laboratory results were available.